# Synthetic Lethal Targeting of Mitotic Checkpoints in HPV-Negative Head and Neck Cancer

**DOI:** 10.3390/cancers12020306

**Published:** 2020-01-28

**Authors:** Alexander Y. Deneka, Margret B. Einarson, John Bennett, Anna S. Nikonova, Mohamed Elmekawy, Yan Zhou, Jong Woo Lee, Barbara A. Burtness, Erica A. Golemis

**Affiliations:** 1Molecular Therapeutics Program, Fox Chase Cancer Center, Philadelphia, PA 19111, USA; alexander.deneka@fccc.edu (A.Y.D.); margret.einarson@fccc.edu (M.B.E.); BennettJ@chc.edu (J.B.); Anna.Nikonova@fccc.edu (A.S.N.); mohamed.elmekawy@fccc.edu (M.E.); 2Department of Biochemistry and Biotechnology, Kazan Federal University, 420000 Kazan, Russia; 3Department of Biology, Chestnut Hill College, Philadelphia, PA 19118, USA; 4Moscow Institute of Physics and Technology, 141701 Dolgoprudny, Russia; 5Bioinformatics and Biostatistics Facility, Fox Chase Cancer Center, Philadelphia, PA 19111, USA; Yan.Zhou@fccc.edu; 6Section of Medical Oncology, Department of Internal Medicine and Yale Cancer Center, Yale University School of Medicine, New Haven, CT 06510, USA; jongwoo.lee@yale.edu (J.W.L.); barbara.burtness@yale.edu (B.A.B.)

**Keywords:** prexasertib, adavosertib, mitotic catastrophe, head and neck cancer

## Abstract

Head and neck squamous cell carcinomas (HNSCC) affect more than 800,000 people annually worldwide, causing over 15,000 deaths in the US. Among HNSCC cancers, human papillomavirus (HPV)-negative HNSCC has the worst outcome, motivating efforts to improve therapy for this disease. The most common mutational events in HPV-negative HNSCC are inactivation of the tumor suppressors *TP53* (>85%) and *CDKN2A* (>57%), which significantly impairs G1/S checkpoints, causing reliance on other cell cycle checkpoints to repair ongoing replication damage. We evaluated a panel of cell cycle-targeting clinical agents in a group of HNSCC cell lines to identify a subset of drugs with single-agent activity in reducing cell viability. Subsequent analyses demonstrated potent combination activity between the CHK1/2 inhibitor LY2606268 (prexasertib), which eliminates a G2 checkpoint, and the WEE1 inhibitor AZD1775 (adavosertib), which promotes M-phase entry, in induction of DNA damage, mitotic catastrophe, and apoptosis, and reduction of anchorage independent growth and clonogenic capacity. These phenotypes were accompanied by more significantly reduced activation of CHK1 and its paralog CHK2, and enhanced CDK1 activation, eliminating breaks on the mitotic entry of cells with DNA damage. These data suggest the potential value of dual inhibition of CHK1 and WEE1 in tumors with compromised G1/S checkpoints.

## 1. Introduction

Head and neck squamous cell carcinoma (HNSCC) affects over 800,000 people annually worldwide, with an estimated 65,000 new cases diagnosed and nearly 15,000 deaths in the U.S. [1]. For HPV-negative HNSCC, about 50% of patients with locally advanced and almost all with distant disease succumb to this cancer. Therapeutic options have expanded recently, but even so, the majority of patients experience progression after chemotherapy, immune checkpoint inhibition, or the combination [2]. Notably, the most common genetic lesion in HPV-negative HNSCC is mutation of the tumor suppressor *TP53* [3], which occurs in >85% of tumors and disrupts a central component of the G1/S checkpoint machinery. Abnormalities of a second tumor suppressor, *CDKN2A*, also involved in G1 arrest, are described in 57% of cases. As a result, many HNSCC display reduced G1/S checkpoint-induced cell cycle arrest in response to DNA-damaging therapies such as irradiation or the platinating compounds that have been the backbone of treatment for HNSCC [4]. These cancers are also commensurately more dependent on non-G1/S checkpoints for monitoring and removal of cells with excessive DNA damage either from therapy [5] or from innate processes such as replication stress [6]. For drugs targeting other phases of the cell cycle, this enhanced dependence potentially provides an increased selectivity for cytotoxicity in G1/S-compromised tumor tissues, but without commensurate cytotoxicity in non-tumor tissue where G1/S checkpoints are intact.

A number of groups have investigated the activity of drugs targeting cell cycle transitions in HNSCC, and other tumors. One strategy has been to inhibit the enzymes required for cell cycle advancement. For example, CDK4/6 kinase inhibitors are activated by association with cyclins D and E, are required for progression through G1 and S phase, have established efficacy against breast cancer, and are currently under evaluation for HNSCC [7,8,9]. Other kinases that have been evaluated as targets in preclinical or clinical studies in HNSCC include Aurora A (AURKA) [10,11,12] and Polo-like kinase 1 (PLK1) [13], required for entry into and progression through M phase. Mechanistically, CDK1 activation proximally governs entry into M phase; CDK1 activation depends on the removal of an inhibitory phosphorylation of Y^15^, which is mediated by the CDC25 family phosphatases. AURKA phosphorylation of CDC25b activates this phosphatase [14]. In parallel, AURKA phosphorylation activates the PLK1 kinase [15,16]. Activated PLK1 phosphorylates and inhibits the WEE1 kinase, responsible for introducing the inhibitory Y^15^ phosphorylation on CDK1 [17,18], and also phosphorylates and activates the CDC25 phosphatases [19], ensuring removal of existing Y^15^ phosphorylations. While inhibitors of these proteins have some clinical activity, single-agent potency is often limited in solid tumors and associated with cytostatic rather than cytotoxic effects [20]. 

Another cell cycle-targeting strategy has been to eliminate cell cycle brakes. Potential therapeutic targets of such a strategy include WEE1—which regulates CDK1—as well as ATR, ATM, CHK1, CHK2 checkpoint kinases that act at multiple phases of the cell cycle to induce arrest in response to DNA damage and other stressors. These drugs induce lethality by driving cells through cell cycle phases in the absence of the ability to correct DNA damage, nutrient imbalance, or chromosome misalignment [21]. This approach, in particular the combination of such checkpoint inhibitors with irradiation, genotoxic drugs, or targeted inhibitors of cell cycle kinases such as AURKA, has been promising in preclinical studies [6,12]. There is clear potential to identify further productive combinations. 

## 2. Results

### 2.1. Profiling Vulnerability to Cell Cycle-Targeted Compounds in Head and Neck Cancer Cell Models

To evaluate the response of p53-compromised HNSCC cells to cell cycle-targeted inhibitors, we used a panel of three well-characterized HNSCC cell lines: CAL27, FaDu, and SCC61. For CAL27 and FaDu, exome sequence information was available in cBioPortal [22] and the Cancer Cell Line Encyclopedia at the Broad Institute [23]; for SCC61, we performed exome sequencing for the purpose of this study. All three models have defined damaging mutations impairing *TP53* function; CAL27 and FaDu additionally have damaging mutations in *CDKN2A* (Table 1). In addition, we included the salivary gland-derived A-253 cell line, which also bears damaging *TP53* and *CDKN2A* mutations as a reference model, to determine if observed responses were specific to HNSCC cells or would also be observed in G1/S checkpoint-deficient cell lines of distinct lineages (Table 1 and Appendix A). We then developed a panel of drugs targeting proteins active in cell cycle regulation (Appendix A and Figure 1a), arrayed with the microtubule-targeting cytotoxic compound docetaxel as a control. Targets included proteins promoting cell cycle progression (CDK4/6, PLK1, AURKA, AURKB) or DNA damage responses (PARP, topoisomerase); or involved in checkpoints active at mitosis (MPS1), G2/M (WEE1), or multiple phases of the cell cycle (CHK1, CHK2, ATM, ATR).

For each of the four cell lines, we used CellTiterBlue assay to compare the reduction in viability of cells induced by each of the drugs on the panel at a single concentration (Figure 1b). While sensitivity differed between cell models, at least two of the three HNSCC models were sensitive (>60% inhibition at 500 nM) to a group of six drugs including adavosertib (WEE1), alisertib (AURKA), danusertib (AURKA and AURKB), prexasertib (CHK1), reversine (MPS1), and BI 2536 (PLK1). We did not further analyze BI 2536 as its reported [25] ability to target BRD4, a regulator of transcriptional elongation, would complicate interpretation of cell cycle phenotypes. For each of the other five drugs, we performed IC50 determination (Figure 1c and Appendix A), which confirmed all the HNSCC cell lines had intermediate or high sensitivity to at least two of the drugs, and the A-253 model was sensitive to prexasertib.

### 2.2. Potent Combination Activity of Prexasertib and Adavosertib

The pattern of drug sensitivity suggested a vulnerability to targeting the G2 and M phases of cell cycle; however, each of the drugs had a distinct target relevant to these cell cycle transitions. We further explored whether the combined application of pairs among the five drugs identified in primary screening would reveal enhanced activity. To do so, we explored the effectiveness of drug combinations at multiple ratios, at concentration ranges below 500 nM. Based on this analysis, 72 hours’ exposure to a combination of prexasertib:adavosertib (pre:ada) was highly effective in reducing cell viability at a 1:10 ratio (25:250 nM), in FaDu, SCC61, and A-253 cells (Figure 2a). CAL27 cells were more resistant to G2/M-targeting drugs used as single agents, which may reflect the fact this cell line contains a damaging mutation in *CASP8* in addition to mutations in *TP53* and *CDKN2A*. For this cell line, the pre:ada combination was most effective when the drugs were used at ratios from 1:2 to 2:1 (100:200 and 200:100 nM). Analysis of combination index for pre:ada revealed synergy in the FaDu cell line, and strong synergy in CAL27 cells, particularly at low concentrations of drug combination (Figure 2b). In contrast, reduction in viability was significantly lower in non-transformed normal human tracheobronchial epithelial cells (NHTBE cells) and the human fibroblast cell model IMR90 (Appendix A).

To evaluate whether drug combinations enhanced cytotoxicity rather than promoting cytostasis, we applied additional complementary approaches. First, we treated the four-cell models with the optimal drug combination ratios for 72 h. We then treated unfixed cells with Hoechst to visualize nuclei in all cells and propidium iodide (PI), which is excluded by viable cells, as an indicator of cell death. Automated imaging confirmed a significant decrease in numbers of cells (Figure 2c) and an increase in cell death (Figure 2d), in CAL27 and FaDu cells treated with the combination versus single agents across a range of concentrations at the selected ratios; this paralleled a decrease and metabolic activity of cells. In contrast, the combination did not show a beneficial effect in relationship to single-agent prexasertib in reducing cell number or increasing cell death in the other two cell models in spite of reducing the metabolic activity of these models (Figure 2c,d; all assays benchmarked at a single concentration in Figure 2e). 

As one possible reason for the discordance between assays, we considered the fact that both prexasertib and adavosertib induced cytostasis, but different cell models have varying ability to overcome cell cycle defects arising from inhibition of CHK1 or WEE1, such that the time of progression from arrest to cell death varies between cell lines. Thus, cytotoxicity may not fully manifest within 72 h in cells grown under adherent conditions. To better assess the physiological effect of exposure to the pre:ada combination, we first evaluated the effectiveness of the pre:ada combination in an anchorage-independent setting, monitoring reduction in viability of 3D spheres following 72 h of treatment with single drugs or the drug combination (Figure 3a). In this setting, the combination was again significantly more effective than single drugs in the FaDu and A-253 models, but not the other two models, suggesting cell adhesion did not have a significant role in influencing drug activity.

To assess the consequences of long-term exposure, we next performed clonogenic assays in each of the cell models. Over the course of 12 days incubation, cells were either treated with fresh drug with each media change (Figure 3b) or only dosed a single time at initial plating (Figure 3c). Drugs were used at the concentrations of 25 nM prexasertib and 250 nM adavosertib for all cell lines except CAL27, where prexasertib was used at a concentration of 500 nM because of intrinsic resistance. In this assay, the impact of the drug combination on viability was extremely marked, resulting between 14–933 times greater reduction in colony formation than each drug used alone, and 2–4 orders of magnitude in colony formation versus vehicle-treated cells. This very striking result implied that dual targeting of WEE1 and CHK1 imposed an insuperable cell cycle defect that required >72 h to fully manifest, and that a single dose of drug was sufficient to induce this defect. In contrast, the degree of growth reduction in IMR90 cells after 12 days was less than one order of magnitude, and no combination effect of the drugs was seen (Appendix A).

### 2.3. Combination of Prexasertib and Adavosertib Enhances DNA Breaks and Causes Mitotic Catastrophe

G2/M checkpoint kinases, including CHK1 and WEE1, negatively regulate entry into M phase [26]. As adavosertib eliminates the WEE1 G2/M checkpoint, and prexasertib eliminates CHK1 activity in S phase and at G2/M, cells treated with both drugs have reduced capacity to initiate cell cycle arrest; they are thus less able to repair ongoing replication-associated DNA damage. Such an effect might explain the markedly greater activity of the drug combination in the longer-term clonogenic assays than the shorter cell viability assays. To address this possibility, we analyzed levels of γ-H2AX, a phosphorylated histone mark associated with unrepaired DNA breaks [27], in cells treated with vehicle, single drugs, or the pre:ada combinations for 72 h (Figure 4a and Appendix A). This showed a significant increase in phospho-H2AX signal in cells treated with the drug combination versus single drugs, even at low overall drug concentrations.

We then treated cells with drugs in either a short-term experiment (48 h; Figure 4b and Appendix A) or in a longer-term experiment matching the time-course of the clonogenic assays (12 days; Figure 4c and Appendix A) to assess alterations in the cell cycle. In CAL27 cells, at 48 h, the combination resulted in considerable accumulation of cells with 4N (G2/M) and >4N DNA, suggesting a mitotic defect. In FaDu cells, by contrast, while some residual viable cells accumulated as >4N at 48 h, in the majority of the population, the drug combination resulted in a high level of cell death, likely as a consequence of failed mitosis (Figure 4d). After 12 days of drug treatment, very few cells remained viable in any cell model. Among those remaining, the cell cycle distribution was abnormal in all surviving drug-treated cells, with evidence for accumulation of aneuploid populations; however, cells treated with the drug combination typically no longer had any detectable 4N DNA peak, suggesting clearance of mitotic cells.

To directly test this possibility, we performed live cell imaging of CAL27 and FaDu cells plated at low density to optimize proliferation rates, synchronized the cells with a double thymidine block, and then treated them with prexasertib and adavosertib alone or in combination (Figure 5). Beginning 2–3 h after the addition of drugs, cells were imaged for 24 h to evaluate the frequency of cells undergoing mitosis and/or cell death during this period. For these cell lines, single drugs or drug combinations had no statistically significant effect on the percentage of dead cells in the overall population, in the absence of prior entry into mitosis (Figure 5a). In addition, the percentage of cells initiating mitosis was not significantly affected (Figure 5b). In contrast, the frequency of cells with evidence of failed mitosis (defined as rounding and progression to metaphase, but then inability to progress through cytokinesis (Figure 5c)), or undergoing mitotic catastrophe (defined as loss of cell integrity during the mitotic process), ultimately resulting in apoptosis, (Figure 5d), was modestly elevated by treatment with either drug alone (representative images in Figure 5d). The frequency of mitotic cell death was significantly enhanced by a pre:ada combination versus the use of single drugs, rising to 50%–70% of total mitoses (Figure 5d,e). Confocal imaging of cells 24 h after addition of drugs confirmed a high frequency of cells with abnormal mitotic spindles in all drug-treated populations (Figure 5f); however, because many of the cells treated with the drug combination underwent apoptosis in mitosis, associated with loss of attachment from the imaged slide, no increase in frequency of abnormal mitotic figures was associated with application of the drug combination.

### 2.4. Combined Application of Prexasertib and Adavosertib Enhances CDK1 Activation

Our data suggested that cells treated with the pre:ada combination would be more likely to enter mitosis due to the elimination of CHK1 and WEE1 checkpoint activity. To test this idea, we treated FaDu, CAL27, and SCC61 cells for 6 or 72 h with vehicle, single drugs, or the drug combination, and used Western analysis to examine expression and activity state for relevant cell cycle regulatory targets (Figure 6). Notably, Y^15^-phosphorylated CDK1, associated with inhibition of CDK1 kinase activity, was very significantly decreased by the pre:ada drug combination at both timepoints, indicating removal of the cellular brakes on kinase activation. In the FaDu and SCC61 cell lines, neither single drug treatment had any effect on CDK1 activity at the 6 h time point. CDK1 activity remained elevated through 72 h of drug treatment, although total levels of CDK1 were reduced in combination-treated cells, likely reflecting high levels of cell death.

We also examined the effect of single and combination drug treatment on CHK1 activity. At 6 h, expression of S^296^-phosphorylated activated CHK1 in the CAL27 cell line and total CHK1 FaDu cells were both reduced by treatment with the pre:ada combination, to a greater extent than seen with prexasertib alone (Figure 6 and Appendix A). We note that it has previously been reported that inhibition of CHK1 is associated with increased proteasomal degradation of the protein in some cell settings [28], explaining the change in total protein expression. Interestingly, although no consistent reduction in CHK1 was observed in the SCC61 cell model, where baseline levels of CHK1 were very low, the activity of CHK2 (T^68^-phosphorylated CHK2) was rapidly induced by the drug combination in all three cell models, implying rapid increase in damage induced by the drug combination (Figure 6).

## 3. Discussion

The data presented here support several conclusions. First, they demonstrate that combined application of adavosertib, a drug inhibiting WEE1, and prexasertib, a drug targeting CHK1, is highly cytotoxic in four *TP53* and *CDKN2A*-mutated HNSCC cell lines. Second, they demonstrate that the cytotoxic effect of the drugs is linked to cell cycle defects that manifest in M-phase cells, causing a mitotic catastrophe. Third, the cytotoxicity and mitotic catastrophe induced by combination drug treatment occur in the context of increased DNA damage signaling (including elevated γ-H2AX), suggesting failure to repair intrinsic DNA damage arising from such processes as replication stress. Fourth, the pre:ada combination causes greater activation of the CDK1 mitotic kinase than either drug used alone and a much greater activation than observed in vehicle-treated cells, suggesting *TP53/CDKN2A*-deficient cancer cells normally require the combined activity of WEE1 and CHK1 to restrain M-phase entry as part of maintaining genomic integrity. Fifth, the pre:ada combination has limited potency in untransformed bronchial epithelial cells and IMR90 human fibroblasts, in comparison to HNSCC cell lines bearing *TP53* and *CDKN2A* mutations. Together, these findings suggest that further preclinical investigation of pre:ada combination is merited. 

The dependence of G1/S checkpoint-compromised tumors on intact regulation of G2/M nominates enzymes controlling the G2/M transition as promising candidates for inhibition, with high specificity for cancer cells. The activities of CHK1 and WEE1 are interconnected in regulating cell cycle. Following checkpoint activation, CHK1 phosphorylation activates WEE1. In turn, WEE1 phosphorylates cyclin-dependent kinase 1 (CDK1), the proximal kinase for mitotic entry, on tyrosine Y^15^, inhibiting its activity (Figure 1). WEE1 is upregulated in the setting of DNA damage (common in *TP53^mut^* HNSCC tumors) as part of the G2/M checkpoint, and *TP53^mut^* HNSCC are sensitive to WEE1 inhibition [29]; WEE1 activity both prolongs S phase [30] and delays the G2/M transition to allow DNA repair [31]. Therefore, WEE1 has been considered as a therapeutic target, with the inhibitor adavosertib (AZD1775) in clinical trials [32,33,34] (NCT02610075). Pre-clinical and clinical data show that WEE1 inhibition leads to DNA damage and accelerated mitotic entry [33,35,36,37]. Adavosertib synergizes with cisplatin in HNSCC models [38] and was notably active in HNSCC in a pre-operative chemotherapy combination trial [34]. CHK1 has also been considered a promising drug target, with a number of agents in pre-clinical and clinical development for use as monotherapy or in combination with DNA damaging agents; among a number of CHK1 targeting drugs, the inhibitor prexasertib has shown particularly good efficacy across multiple cancer types [39,40]. However, no clinical assessment of combined prexasertib and adavosertib treatment has been performed in HNSCC or any other solid tumor. Our data suggest this drug combination may represent a promising modality for treating HPV-negative HNSCC and other *TP53*-mutated cancers.

A notable finding of our work is the profound impact of the pre:ada combination on clonogenic survival following a single exposure to low dose combination treatment. Taken together with the minimal toxicity we observed in normal epithelial cells and IMR90 fibroblasts, this offers the possibility that combination therapy can be administered clinically with a suitable therapeutic window. Early phase single-agent clinical trials with prexasertib and adavosertib have indicated the potential for activity—including in HNSCC—as well as significant myelotoxicity. In a phase II trial, the recommended dose and schedule for prexasertib monotherapy was established as 105 mg/m^2^ once every 14 days [41]. In this trial, the incidence of grade 4 neutropenia was over 70%, and an objective response was observed in a patient with HNSCC. Objective responses and disease control in HNSCC were confirmed in a single-agent expansion cohort [42]. Similarly, an initial observation of modest single-agent activity for adavosertib in HNSCC was followed by an induction study incorporating adavosertib with chemotherapy and demonstrating activity in the pre-operative setting [34,43]. 

While preclinical evidence exists for a synergistic combination effect between inhibitors of WEE1 or CHK1 with PARP [44,45], AURKA [12], and EGFR inhibitors, as well as with standard cytotoxic approaches, few clinical trials of combinations have moved forward because of concerns about additive toxicity. Those combination studies completed indicate that neutropenia is dose-limiting for combination therapy, as predicted [46]. The clonogenic survival data presented here indicate that rationally designed combinations accelerate mitotic cell death in p53- and CDKN2A-deficient cells, but not normal tracheobronchial epithelia or human fibroblasts. Given similar tolerability of this combination by myeloid precursors, these results suggest that combined use of pre:ada will reduce the post-treatment survival of populations of resting G1/S-compromised HNSCC cells, and do so at dramatically lower doses than required for single-agent activity. Importantly, this provides an avenue forward for such combinations despite an overlapping myelotoxicity profile at the doses required for single-agent activity. 

Future studies will be needed to evaluate the in vivo efficacy versus tolerability of the pre:ada drug combination, including the use of xenograft and patient-derived xenograft (PDX) models, coupled with the establishment of biomarkers for optimal response. Such biomarkers would include the somatic mutation profile of tumors, with emphasis on *TP53* and *CDKN2A* mutational status, among others. In addition, the primary screen in this study identified a number of other drugs that would be useful to evaluate in future combination studies. These would include AMG900, a pan-AURKA inhibitor; BI 2536, a PLK1/BRD4 inhibitor; and reversine, which combines pan-AURKA inhibition with additional inhibition of MPS1. Based on the current data, we anticipate that combinations of such agents may increase anti-tumor potency without enhancing non-specific cytotoxicity, potentially benefiting HNSCC patients. 

## 4. Materials and Methods 

Cell Culture. The CAL27 (ATCC CRL-2095), FaDu (ATCC HTB-43), and SCC61 (PMID: 3182330) HNSCC and A-253 submaxillary salivary gland carcinoma cell lines were obtained from the FCCC Cell Culture Facility and authenticated by genotyping performed by IDEXX BioResearch (Columbia, MO, USA). The SCC61 and FaDu cell lines were cultured in DMEM-F12 media containing 20% fetal bovine serum (FBS), L-glutamine (L-glu), 0.4 µg/mL hydrocortisone, and penicillin/streptomycin (pen/strep). CAL27 and A-253 (ATCC HTB-41) were cultured in DMEM media containing 10% FBS, L-glu and pen/strep. NHTBE cells (purchased from Lonza, Durham, NC, USA) were maintained in bronchial epithelial cell growth medium (BEGM) supplemented with BEGM BulletKit (Lonza, CC-3170), as previously described [12]. IMR90 (ATCC CCL-186) human normal lung fibroblasts were purchased from ATCC and cultured in DMEM medium containing 10% FBS, sodium pyruvate, L-glu, non-essential amino acids, and pen/strep. 

Sequencing of SCC61. SCC61 cells had been previously described as *TP53* mutant. We undertook whole exome sequencing (WES) to confirm this; WES was performed by the Yale Center for Genome Analysis, as previously described [47]. Fastq files from targeted sequence capture were processed in an exome analysis pipeline. Reads were aligned to human genome reference hg19 with BWA [48]. PCR duplicates were removed using Picard tools (v2.1.1) and GATK (v.3.6-0) pipeline was used for quality score recalibration, realignment, and variant calling [49]. The resulting VCF file was annotated in Annovar [50] based on the presence of common variants in the ExAC database and 1000Genomes project. Annovar software was also used to predict mutations affecting protein function using pathogenicity prediction score values generated by algorithms including SIFT, M-CAP, MetaLR, Polyphen-2, LRT, MutationTaster, FATHMM, PROVEAN, VEST3, and MetaSVM. Sequences have been submitted to the NCBI Sequence Read Archive (SRA), accession number PRJNA591574.

Cell proliferation and viability assays and calculation of drug parameters. To analyze the effects of drug treatment on proliferation, cells (2000 cells/well) were plated in 96-well cell culture plates in complete media. After 24 h, vehicle and compounds at the concentrations indicated were added; 500 nM was chosen as an initial screening concentration to increase the likelihood of identifying compounds with high in vivo potency. Viability assay measurements were performed at 96 h after drug addition using CellTiterBlue reagent (#G808A, Promega, Madison, WI, USA) according to the manufacturer’s protocols. In the initial high throughput screen, drugs were added using a CyBio Well Vario 96/384 channel automated pipettor (CyBio Inc, Woburn, MA, USA). All assays were performed in 3 technical repeats and 3 biological repeats. Data for drug treatment was normalized to vehicle conditions and processed in Microsoft Excel software. For single-agent analysis, we estimated the IC50 using GraphPad Prism 7 with multiple dose-response models. 

Automated immunofluorescent live/dead cell detection. The quantitative image-based assay detecting live/dead cell was performed in triplicate in a 96-well format. Twenty-four hours after plating, cells were treated with indicated compounds and vehicle. After 72 h incubation, Hoechst 33342 dye (final concentration 5 μg/mL; H3570, Thermo Scientific, Waltham, MA, USA) and propidium iodide (PI; final concentration 0.5 μg/mL; R37108, Thermo Scientific, Waltham, MA, USA) were added to the cells according to manufacturer protocols and incubated for 30 min. Live cell imaging was performed with an automated high-throughput screening–microscope (ImageXpress Micro, Molecular Device Sunnyvale, CA, USA), driven by MetaXpress software (Molecular Devices, Sunnyvale, CA, USA); four independent 4× image fields from each individual well were acquired. Images were segmented for analysis using the “Multiwavelength Scoring” module (MetaXpress, Molecular Devices), using Hoechst for nuclear segmentation, and scoring PI positivity for cell death. Results from these analyses were displayed within Acuity (Molecular Devices, Sunnyvale, CA, USA) and further processed in Microsoft Excel and GraphPad Prism (GraphPad Software, San Diego, CA, USA).

Automated immunofluorescence detection of γ-H2AX. The quantitative image-based assay of foci containing phosphorylated histone H2AX (γ-H2AX) was performed in triplicate in 96-well plates. Cells were treated with vehicle, prexasertib, and AZD1775 in concentrations indicated, 24 h after plating. After 72 h incubation, cells were fixed, stained with anti-γ-H2AX primary antibodies (1:1000, Mouse Monoclonal, Millipore Upstate, Billerica, MA, USA), followed by staining with FITC-tagged secondary antibodies (1:1000, goat anti-mouse IgG (H+L), Alexa Fluor® 488 conjugate), and Hoechst 33342 nuclear stain (H3570, Thermo Scientific, Waltham, MA, USA), and imaged using an ImageXpress micro automated microscope and scored as previously described [51]. Results from these analyses were displayed within Acuity (Molecular Devices) and further processed in Microsoft Excel and GraphPad Prism.

Confocal immunofluorescence microscopy. Immunofluorescence was used to detect mitotic spindle abnormalities, using standard protocols. Cells were pretreated with indicated drugs or vehicle for 24 h. Antibodies to acetylated α-Tubulin (6-11B-1, Santa Cruz, Dallas, TX, USA) and phospho-Histone H3 (#3377S, Cell Signaling) were used as primary detection reagents, and secondary Alexafluor 568-tagged donkey anti-rabbit and Alexafluor 488 goat anti-chicken (Life Technologies, Eugene, OR, USA) as secondary antibodies. Slides were counterstained with Vectashield mounting medium with DAPI (4′, 6-diamidino-2-phenylindole) (H-1200, Vector Labs, Burlingame, CA, USA) solution to visualize DNA. Samples were imaged at room temperature (RT) using an SP8 confocal system equipped with an oil immersion 363 objective with numerical aperture (NA) 1.4 (Leica Microsystems, Buffalo Grove, IL, USA) and LASAF (Leica Application Suite Advanced Fluorescence, Buffalo Grove, IL, USA) software. 

Live cell imaging. Time-lapse multi-field experiments were performed in phase contrast on an automated inverted Nikon Eclipse TE300 microscope equipped with thermal and CO_2_ regulation (Nikon, Melville, NY, USA). Two days prior to imaging, 60 × 10^3^ cells/well were plated on a 12-well plate, and synchronized using double thymidine block with 2 mM thymidine. Immediately after cells were released from the second block, compounds were added and cells were imaged at 15-min intervals with a 10× objective and an EZ CoolSnap CCD camera (Roper Scientific/Teledyne Photometrics, Tucson, AZ, USA) for 24 h, and then stacked into movies using Metamorph (Universal Imaging Corps, West Chester, PA, USA) software. The videos were analyzed using ImageJ software (http://imagej.nih.gov/ij/docs/guide/146.html). Cells in every field were annotated and mitotic events were counted. Data were processed in Microsoft Excel software and statistical differences were calculated for each condition using GraphPad Prism software. 

Clonogenic survival assays. In 6-well plates, 500–1000 cells were plated and incubated in complete media. After 24 h, indicated drugs and vehicle were added. Medium was replaced every three days. In one arm of the experiment, cells were incubated for 12 days without further drug replenishment; in a second arm, drugs were refreshed with medium change. Cells were fixed in 10% acetic acid/ 10% methanol solution and stained with 0.5% (*w/v*) crystal violet, as previously described [52]. A colony was defined as consisting of >50 cells and counted digitally using ImageJ software, as described previously [53].

Anchorage-independent growth assays. Three thousand cells/well were seeded in complete medium in CellStar 96-well u-bottom cell-repellent surface plates (Greiner Bio-One GmbH, Kremsmünster, Austria). After 24 h, drugs were added and cells incubated for 72 h. Cell viability was assessed by a CellTiterGlo luminescent cell viability assay (#G7570, Promega, Madison, WI, USA), using the manufacturer’s protocols. Data were processed in Excel software and statistical differences were calculated for each condition using GraphPad Prism software. 

Flow cytometry assays for cell cycle analysis. Cells were treated with indicated drugs and vehicle for 48 h and 12 days, stained with PI/RNase staining buffer (BD Biosciences, San Jose, CA, USA). Cell cycle distribution was acquired by flow cytometry on the BD LSRII Flow cytometer and analyzed with FlowJo software (FlowJo LLC, Ashland, OR, USA).

Western blot analysis. HNSCC cells were treated with indicated drugs for 6 and 72 h and lysed in CelLytic MT Cell Lysis Reagent (Sigma-Aldrich, St. Louis, MO, USA). Protein concentrations of the resulting lysates were measured using the Pierce BCA Protein Assay Kit (Thermo Scientific, Waltham, MA, USA). Western blotting was performed using standard procedures, and blots developed by chemiluminescence using Luminata Western HRP substrates (Classico, Crescendo and Forte, EMD Millipore, Burlington, MA, USA). Primary antibodies were used at 1:1000 dilution, and included anti- pS^296^-CHK1 (#2349S), CHK1 (#4696S), CHK2 (#3440S), pY^15^-CDK1 (#9111L), CDK1 (#9116S) (all from Cell Signaling, Danvers, MA, USA), and pT^68^-CHK2 (#AF1626; from R&D Systems Inc., Minneapolis, MN, USA). Secondary anti-mouse and anti-rabbit HRP-conjugated antibodies (GE Healthcare, Little Chalfont, UK) were used at a dilution of 1:10,000. Quantification of signals on Western blots was done using NIH ImageJ software with signaling intensity normalized to loading control.

Statistical analysis. Plots and bar graphs depict the mean and standard error of the mean (SEM) as calculated by Student’s *t*-test. Differences between treatment groups were determined by one- or two-way ANOVA followed by Bonferroni’s post-test. Integration of clonogenic survival as a function of dose, or area under the curve, was calculated using GraphPad Prism Software, as were assessments of protein expression. 

## 5. Conclusions

In this study, we screened a panel of cell cycle- and checkpoint-targeted drugs for activity in a panel of cell models with defined defects in *TP53* and *CDKN2A* status; for selected compounds with single-agent activity, we evaluated combination effects in detail. Based on this work, we found that dual inhibition of WEE1 and CHK1 using the clinical compounds adavosertib and very low doses of prexasertib resulted in enhancement of DNA damage, mitotic catastrophe, and control of clonogenic cell growth, associated with elevated activation of CDK1. These data suggest that a continued investigation of this drug combination may be a productive strategy for head and neck cancers and other tumors with compromised G1/S checkpoints.

## Figures and Tables

**Figure 1 cancers-12-00306-f001:**
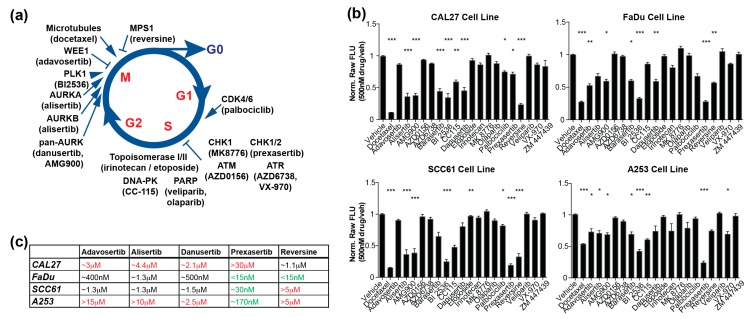
Initial high throughput screening results in multiple HNSCC cell lines using 24 compounds. (**a**) Schematic representation of drugs targeting proteins active in cell cycle regulation; (**b**) Cell viability determination by CellTiterBlue assay for 4 cell lines treated with drugs indicated at 500 nM concentration; (**с**) Summary of IC50 values for selected compounds for four cell lines. Greater resistance, red font; more sensitive, green font. See also Appendix A. All graphs: *, *p* ≤ 0.05; **, *p* ≤ 0.01; ***, *p* ≤ 0.001 relative to vehicle control.

**Figure 2 cancers-12-00306-f002:**
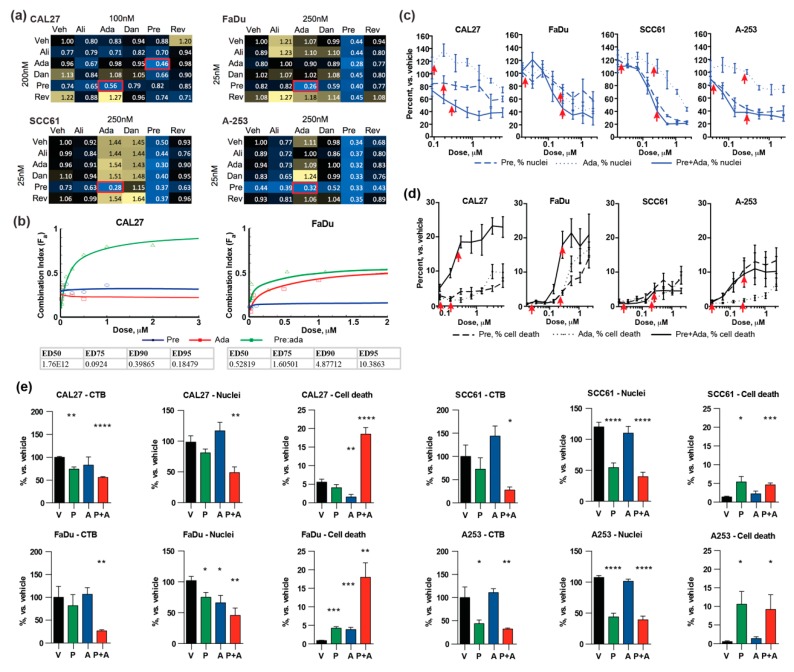
Evaluation of combination effect of selected cell cycle inhibitors. (**a**) Heatmaps representing values from CellTiterBlue (CTB) assays for combinations of the indicated targeted drugs, at the concentrations indicated. Values indicate reduction in cell metabolic activity normalized to vehicle control; blue, positive combination activity; yellow, antagonistic combination activity; red boxes, optimal combination; (**b**) combination index (CI) calculation results for the CAL27 and FaDu cell lines to assess synergistic effects of pre:ada drug combination; (**c**,**d**) dose response curves indicating viable cells (based on nuclear count) (**c**) and inviable cells (based on failure to exclude propidium iodide) (**d**). Arrows indicate concentrations comparable to those shown in CTB assays in (**a**); (**e**) Comparison of results from assays in (**a–d**) at drug concentrations used in (**a**). All graphs: *, *p* ≤ 0.05; **, *p* ≤ 0.01; ***, *p* ≤ 0.001; ****, *p* ≤ 0.0001 relative to vehicle controls.

**Figure 3 cancers-12-00306-f003:**
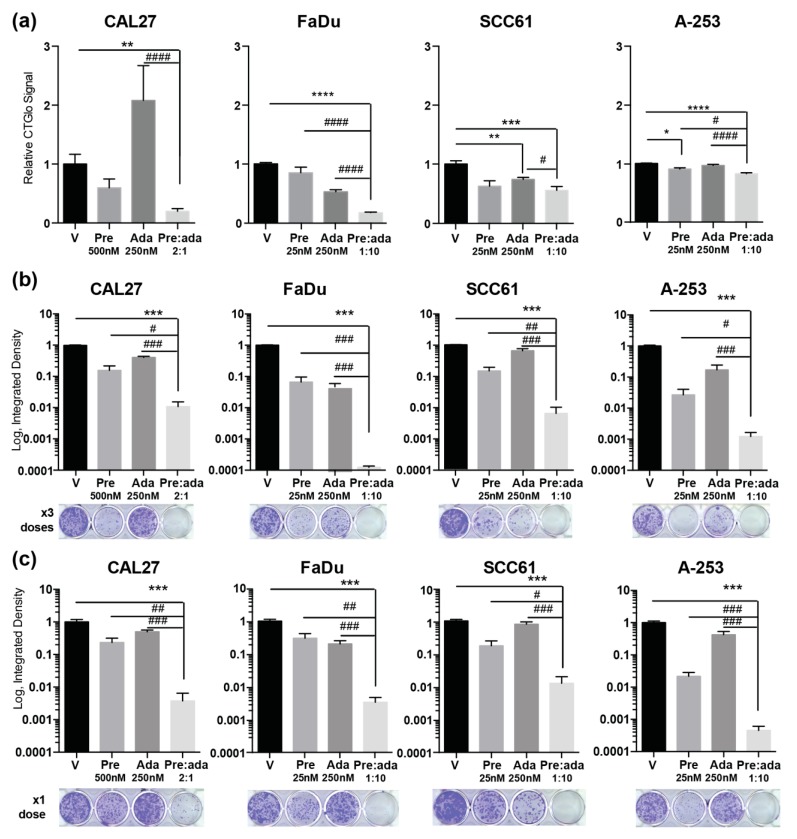
Combination of adavosertib and prexasertib in assays for anchorage-independent growth and clonogenic survival. (**a**) Metabolic activity measured by CellTiterGlo in cells grown in anchorage-independent conditions for 72 h of treatment with vehicle or indicated drugs; (**b**,**c**) quantification and representative images for clonogenic growth after 12 days of treatment with drugs indicated, either followed by replenishment of drugs at 3-day intervals (**b**), or with cells receiving only a single initial dose (**c**). All graphs: * or #, *p* ≤ 0.05; ** or ##, *p* ≤ 0.01; *** or ###, *p* ≤ 0.001, **** or ####, *p* ≤ 0.0001 relative to indicated comparator. We note that NHTBE cells must be grown under specialized conditions, at an air-medium interface, and hence cannot be used for a clonogenic assay; further, because they are highly migratory, IMR90 cells do not form colonies; see Appendix A for further analysis.

**Figure 4 cancers-12-00306-f004:**
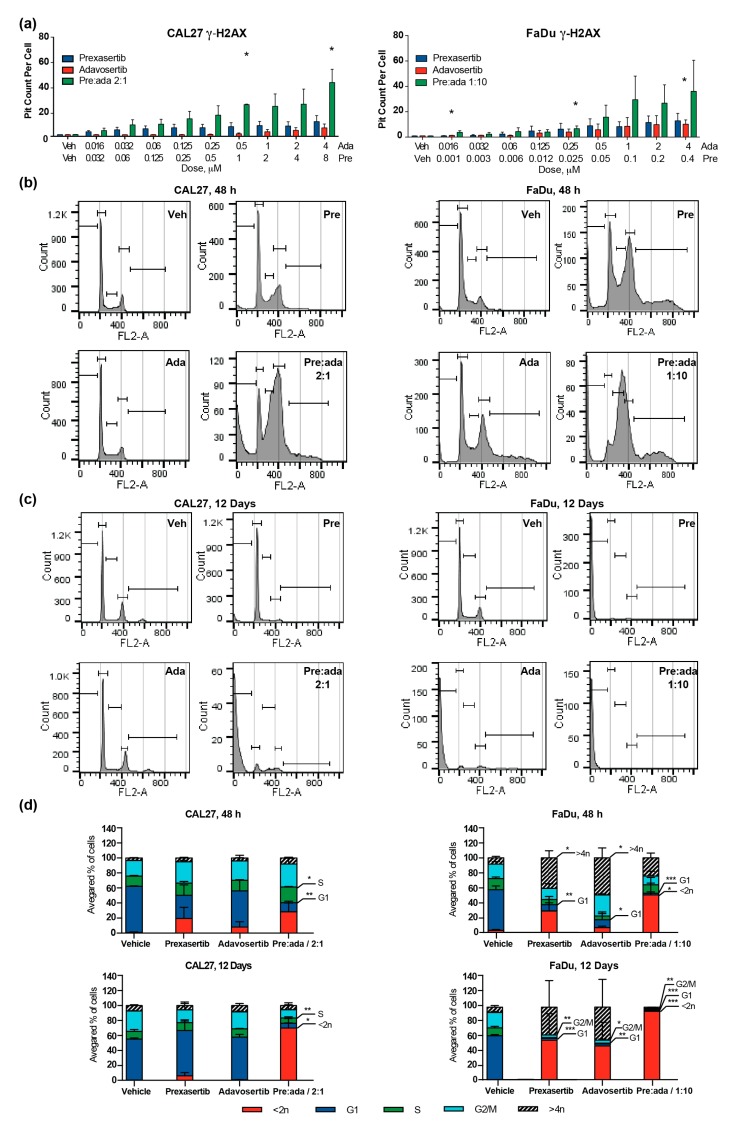
Combination of prexasertib and adavosertib enhances DNA breaks and aneuploidy. (**a**) γ-H2AX-positive foci in CAL27 and FaDu cell lines, 72 h after treatment with vehicle or drugs at concentrations indicated; (**b**,**c**) representative histograms from single runs of FACS analysis of CAL27 and FaDu cells from short term (48 h) (**b**) and long term (12 days) (**с**) treatment with vehicle, or drugs indicated; (**d**) quantification of cell cycle compartmentalization changes, averaged from three repetitions of experiments shown in (**b**), (**c**). All graphs: *, *p* ≤ 0.05; **, *p* ≤ 0.01; ***, *p* ≤ 0.001 relative to vehicle controls.

**Figure 5 cancers-12-00306-f005:**
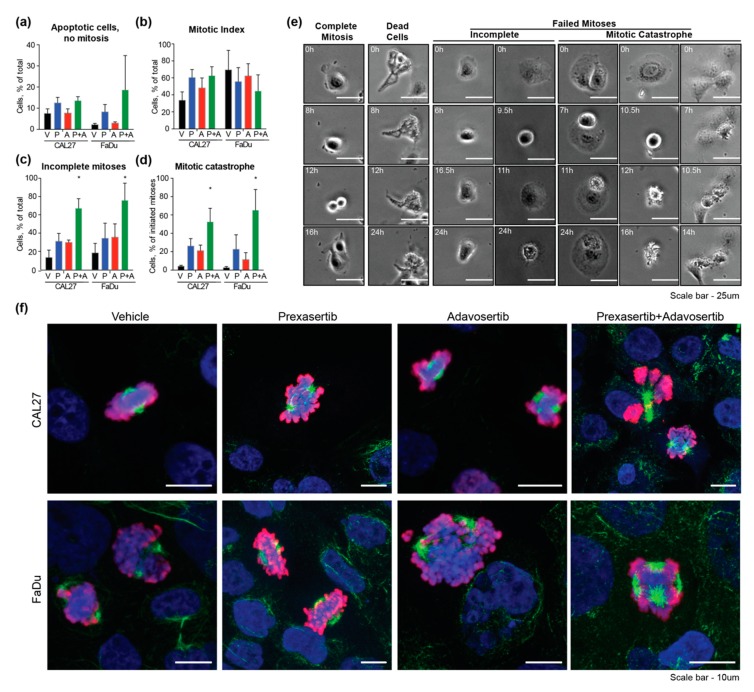
Combination of prexasertib and adavosertib causes mitotic catastrophe. (**a–d**) Quantification of percent of CAL27 and FaDu cells (**a**) undergoing apoptosis without prior mitosis; (**b**) entering into mitosis; (**c**) unable to successfully complete mitosis; (**d**) undergoing apoptosis after initially entering mitosis over 24 h of live cell imaging following addition of indicated drugs; and (**e)** representative images of CAL27 cells related to (**a–d**). Scale bar, 30 um; (**f**) confocal images of cells, stained with DAPI to visualize nuclei (blue), and antibody to acetylated α-tubulin to visualize mitotic spindle (green), or to phospho-histone H3 to visualize DNA breaks (red), 24 h after treatment with indicated drugs. Scale bar, 10 um. All graphs: *, *p* ≤ 0.05; relative to vehicle controls.

**Figure 6 cancers-12-00306-f006:**
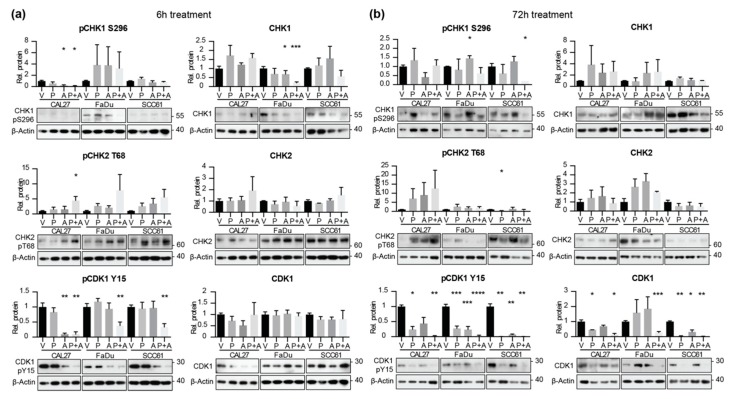
Combined application of prexasertib and adavosertib enhances CDK1 activation. (**a**,**b**) Quantification (top) and representative western blot images (bottom) for indicated phospho- and total proteins in HNSCC cell lines, after 6 (**a**) or 72 h (**b**) of treatment with indicated drugs. All graphs: *, *p* ≤ 0.05; **, *p* ≤ 0.01; ***, *p* ≤ 0.001, ****, *p* ≤ 0.0001 relative to vehicle controls.

**Table 1 cancers-12-00306-t001:** Mutational profile of cell line models. For models shown in the study, mutational profile is provided, in reference to frequency of most commonly mutated genes in HNSCC, based on analysis of HNSCC provisional dataset (as of November 2019). Mutations were called as likely damaging (red) versus likely tolerated or of uncertain significance (black) based on use of pathogenicity prediction scores obtained through Annovar (see Methods) and variant annotations accessible in publicly available databases, including the Cancer Cell Line Encyclopedia (CCLE), gnomAD, ClinVar, and OncoKB. SCC61 contains a polymorphism, P72R, associated with altered TP53 function [24].

Gene	TCGA Provisional	CAL27		FaDu		SCC61		A-253	
	% Samples	AA Change	Type	AA Change	Type	AA Change	Type	AA Change	Type
TP53	72	p.H193L	Missense	p.R116L	Missense	p.R110L	Missense	p.E48Gfs*66	Frameshift del
				NA	Splice site	p.P72R	Polymorphism		
CDKN2A	22	p.E18*	Stopgain	NA	Splice site			p.A17Pfs*4	Frameshift del
CASP8	11	p.II174fs	Frameshift del						
		p.V460V	Silent						
FAT1	23			p.K3277Nfs*3	Frameshift del	p.R2567H	Missense		
HUWE1	9							p.D1193N	Missense
HLA-A	6	NA	Splice site			p.D251H	Missense		
						p.E87K	Missense		
						p.E87D	Missense		
EP300	8					p.I997V	Missense		
						p.M2015I	Missense		
CREBBP	7	p.T2390N	Missense					p.A2008A	Silent
HRAS	6							p.C154C	Silent
AJUBA	6					p.E310Q	Missense		
TGFBR2	5					p.P129Afs*2	Frameshift ins	p.R537P	Missense
HLA-B	5					p.I90M	Missense		
						p.D54G	Missense		
						p.Q94H	Missense		
DYSF	4					p.A171E	Missense		
KDM6A	4	p.E1023K	Missense	p.A15_A17del	In-frame del				
RAC1	2.7	p.L192L	Silent						
PTEN	2.7							p.L152P	Missense
ZNF233	1.8							p.C389R	Missense

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
