# Peer review of "Synthetic Lethal Targeting of Mitotic Checkpoints in HPV-Negative Head and Neck Cancer"

_cancers, 2020, doi:10.3390/cancers12020306_

Round 1

Reviewer 1 Report

The authors specifically analyzed cell cycle regulators, and identified the optimal combination of prexasertib and adavosertive (Pre:ada) using cell lines with TP53 and CDKN2A damaging mutation. They demonstrated  that the low dose Pre:ada combination resulted in enhanced DNA damage, mitotic catastrophe, and reduction of clonogenic cell growth through elevated CDK1 activation. Although the results are derived from in vitro data, the study design and the discussion are reasonable to suggest the merit of continued investigation of Pre:ada combination therapy for head and neck cancer. Some points should be reconsidered.

Line 81, the pathology of A-253 cell line should be described.

Table 1, Polymorphism of SCC61 is written in blue. 

All figures, (a), (b),(c)... are indicated with capital letters in the main text, e.g. Figure 1A, 1B...

Line 128, and Supple Fig S2, is the ratio of Pre:ada 1:10(25:250)? Which dose is indicated for combination graph prexasertib or adavosertib?

Figure 3, The dose of pre and ada are unclear. They should be described in the main text or in the material and method section.

Figure 4A and Supple Fig 3A Right panel, the dose of Pre are correct? 

0.025, 0.05, 0.01, 0.02, 0,04?

Figure 4 B,C,D, Why is the dose of Pre:ada 2:1 for FaDu? 

Line 195, (b) is expressed with wrong character.

Line 200, Supple Fig 3B should be 3C.

Line 300, HPV-HNSCC should be HPV-negative HNSCC?

Figure 5E, Which cell line do the photos represent CAL27 or FaDu?

Author Response

We thank the reviewer for the overall positive assessment of the study.

Comment: Line 81, the pathology of A-253 cell line should be described.

 Response: We have added ATCC information or Pubmed citations for each cell line in the Methods section; these links describe the pathology in detail.

Comment: Table 1, Polymorphism of SCC61 is written in blue. 

 Response: We have corrected the font to black.

Comment: All figures, (a), (b),(c)... are indicated with capital letters in the main text, e.g. Figure 1A, 1B...

Response: We have modified the labeling to lower case letters in the main text.

Comment: Line 128, and Supple Fig S2, is the ratio of Pre:ada 1:10(25:250)? Which dose is indicated for combination graph prexasertib or adavosertib?

Response: We apologize for the confusing presentation of the data.  For the indicated line and figure, the relevant ratio and dose are 1:10 (ranging from 5:50nM to 2:20uM).  This is indicated as “Pre:Ada 1:10” in the graph. We also added this information to the Supplementary Figure S2 caption to make this more clear.

Comment: Figure 3, The dose of pre and ada are unclear. They should be described in the main text or in the material and method section.

 Response: Drugs were used at concentrations 25nM prexasertib and 250nM adavocertib for all cell lines, except CAL27, where prexasertib was used at a concentration of 500nM due to fundamental resistance to this drug. We have added this information to the main text and the Figure 3.

Comment: Figure 4A and Supple Fig 3A Right panel, the dose of Pre are correct?  0.025, 0.05, 0.01, 0.02, 0,04?

Response: We thank the reviewer for catching this error. The correct sequence is 0.025, 0.05, 0.1, 0.2, 0.4; we have adjusted the text accordingly in Fig4A and Supp Fig 3A.

 Comment: Figure 4 B,C,D, Why is the dose of Pre:ada 2:1 for FaDu? 

Response: The correct dose is 1:10, we thank reviewer for catching this typo.

Comment: Line 195, (b) is expressed with wrong character.

Response: We have corrected this.

Comment: Line 200, Supple Fig 3B should be 3C.

Response: We have corrected this.

Comment: Line 300, HPV-HNSCC should be HPV-negative HNSCC?

 Response: Yes, HPV-indicates HPV-negative. We have made this more clear by omitting the superscript and spelling out the designation.

Comment: Figure 5E, Which cell line do the photos represent CAL27 or FaDu?

Response: The cell lines shown represents CAL27 cell line. We have added this information to the Figure Legend.

Reviewer 2 Report

Deneka et al. made a very complex and large investigation about the effects of treatment with various cell cycle inhibiting drugs on HNSCC cell lines with inactivated TP53 and CDKN2A. Their experiments are mostly very well described and stringent. However, some questions still should be answered:

Why are 500 nM used as start concentration for all tested drugs?

Why was AMG900 excluded from further analyses although similar results as for Alisertib could be seen also for this drug.

For better comparison in Fig. 2e graphs should be sorted according to the read-out and more importantly the y-axis must have the same values for all four cell lines for each parameter, i. e. up to 25 % for cell death or either 150 % or 200 % for CTB. This should be also corrected in all other figures where I have not noted it separately.

Fig. 4. B) The reviewer could not follow the authors that after 48 h the G2/M or the >4n amount of cells really increased in CAL27 after combination treatment compared to treatment with only Pre. Also in FaDu combination treatment did not result in higher levels of >4n than in single treated cells, in contrast less of these cells were shown. But in both cases, the trend seemed to be visible which was later after 12 days more prominent that the amount of <2n cells increased and the amount of >4n cells decreased.

Supp Fig S2: Showing only one experiment for non-transformed normal epithelial cells is for sure not sufficient to bolster the statement that the pre:ada combination has limited potency on non-transformed cells (lines 279/280). At least the clonogenic survival must be also done with these cells to prove that combination therapy will not affect normal tissue cells.

It would be interesting to read which further preclinical investigation of pre:ada combination is merited (lines 280/281).

Minor:

Description of Results from Fig 3a in line 161: the authors probably meant CAL27 and FaDu models.

Again in Fig 3 b) and c) same values on y-axis are required. In the actual diagrams on the first glance it looks like single initial dose was more efficient than three doses in SCC61 and A-253.

According to the correct writing of long sequences of digits ISO 31-0 specifies that such groups of digits should never be separated by a comma or point, as these are reserved for use as the decimal sign. So do not write 800,000 people but 800 000 people!

Author Response

We thank the reviewer for the general positive description of the study.

Comment: Why are 500 nM used as start concentration for all tested drugs?

Response: 500 nM was used as a starting concentration because some drugs that are in clinical trial for HNSCC (for instance, alisertib) are effective at or below this concentration in vitro; hence, it seemed a reasonable benchmark. As our goal was to move the combinations rapidly towards in vivo testing, and optimally towards clinical trial, we focused on drugs of relatively high initial potency. We have added commentary on this point to the methods.

Comment: Why was AMG900 excluded from further analyses although similar results as for Alisertib could be seen also for this drug.

Response: We excluded AMG900 from combination studies because of logistical limitations across multiple cell lines. As alisertib was already being assessed as a representative Aurora-A inhibitor at clinically relevant doses, and as AMG900 is a pan-Aurora inhibitor, making it more difficult to explore single effector mechanisms of the drug, alisertib was preferred for combination studies.  We agree that it would be of interest to further explore this drug and  others we did not have capacity to investigate for this study. We have added text to the discussion indicating additional drugs it would be of interest to evaluate.

Comment: For better comparison in Fig. 2e graphs should be sorted according to the read-out and more importantly the y-axis must have the same values for all four cell lines for each parameter, i. e. up to 25 % for cell death or either 150 % or 200 % for CTB. This should be also corrected in all other figures where I have not noted it separately.

Response: We agree that the graphing was not optimal in the first version of the manuscript.  We have redone all the figures following this suggestion (see revised Figures 1b; 2c-e; 3a; 4a, d; S3a), and agree this improves clarity.

Comment: Fig. 4. B) The reviewer could not follow the authors that after 48 h the G2/M or the >4n amount of cells really increased in CAL27 after combination treatment compared to treatment with only Pre. Also in FaDu combination treatment did not result in higher levels of >4n than in single treated cells, in contrast less of these cells were shown. But in both cases, the trend seemed to be visible which was later after 12 days more prominent that the amount of <2n cells increased and the amount of >4n cells decreased.

Response: We regret that we did not adequately explain the data in the original submission.  In 4B, the result reflects a single experiment; 4D represents the average of multiple experiments, which for some conditions was not identical to single runs due to natural experimental variation. The main text describes the result of the averaged experiments. We have also modified the description of the results to agree very clearly with the statement that in either case, by 12 days of growth, cells are largely inviable, with fragmented DNA.

Comment: Supp Fig S2: Showing only one experiment for non-transformed normal epithelial cells is for sure not sufficient to bolster the statement that the pre:ada combination has limited potency on non-transformed cells (lines 279/280). At least the clonogenic survival must be also done with these cells to prove that combination therapy will not affect normal tissue cells.

Response:  We agree that presenting proliferation assays only for NHTBE cells may limit our conclusions.  Unfortunately, this physiological cell line must be grown under specialized conditions, at an air-medium interface, and hence cannot be used for a clonogenic assay. We have made this point clear in the figure legend to the clonogenic assay.

As an alternative approach, we have performed additional experiments using IMR90 human lung fibroblasts. In a proliferation assay, these cells were largely insensitive to the drugs and drug combination at concentrations in excess of those that significantly reduced viability of the HNSCC cell lines, as we now show in new Supp Figure S2b. We also evaluated the IMR90 fibroblast line in a clonogenic assay, as suggested.  This did not yield data directly comparable to that for the HNSCC cell lines, because the cells were highly migratory, and did not form discrete colonies.  However, these data were suitable to report cell survival after 12 days of exposure to the drug, based on quantitation of total cell staining.  We found that at concentrations of drugs that resulted in 2-4 orders of magnitude reduction in the number of colonies of HNSCC cells, we only observed a reduction in the cell mass in IMR90 of 1 order of magnitude or less. These data are now included in new Supp Figures S2c and S2d, support our general idea that the drugs are more effective in TP53-mutated HNSCC cells. We have also modified the discussion to indicate the need for further testing.

Comment: It would be interesting to read which further preclinical investigation of pre:ada combination is merited (lines 280/281).

Response: We have extended the discussion to address specific areas of investigation that would be of interest.

Minor Comments:

Comment: Description of Results from Fig 3a in line 161: the authors probably meant CAL27 and FaDu models.

Response: We indeed meant the FaDu and A253 models, because based on determination of statistical significance, the combination was more effective compared to single drug efficacy (as indicated by # signed comparison on the Figure). 

Comment: Again in Fig 3 b) and c) same values on y-axis are required. In the actual diagrams on the first glance it looks like single initial dose was more efficient than three doses in SCC61 and A253.

Response: We have redrawn the figure to remove any possible confusion.

Comment: According to the correct writing of long sequences of digits ISO 31-0 specifies that such groups of digits should never be separated by a comma or point, as these are reserved for use as the decimal sign. So do not write 800,000 people but 800 000 people!

Response: We have used the standard convention for US English, as enforced by many journals we have published in. If this paper is accepted by Cancers, we will be happy to modify our presentation of numbers to match your requirements.

Round 2

Reviewer 2 Report

The authors followed the reviewer's suggestions.

Author Response

All grammar and spelling have been carefully checked.